# A Microvascular Segmentation Network Based on Pyramidal Attention Mechanism

**DOI:** 10.3390/s24124014

**Published:** 2024-06-20

**Authors:** Hong Zhang, Wei Fang, Jiayun Li

**Affiliations:** School of Information Engineering, Minzu University of China, Beijing 100081, China; weifang475@gmail.com (W.F.); yun1834152@163.com (J.L.)

**Keywords:** vessel segmentation, attention mechanism, U-Net, residual unit

## Abstract

The precise segmentation of retinal vasculature is crucial for the early screening of various eye diseases, such as diabetic retinopathy and hypertensive retinopathy. Given the complex and variable overall structure of retinal vessels and their delicate, minute local features, the accurate extraction of fine vessels and edge pixels remains a technical challenge in the current research. To enhance the ability to extract thin vessels, this paper incorporates a pyramid channel attention module into a U-shaped network. This allows for more effective capture of information at different levels and increased attention to vessel-related channels, thereby improving model performance. Simultaneously, to prevent overfitting, this paper optimizes the standard convolutional block in the U-Net with the pre-activated residual discard convolution block, thus improving the model’s generalization ability. The model is evaluated on three benchmark retinal datasets: DRIVE, CHASE_DB1, and STARE. Experimental results demonstrate that, compared to the baseline model, the proposed model achieves improvements in sensitivity (Sen) scores of 7.12%, 9.65%, and 5.36% on these three datasets, respectively, proving its strong ability to extract fine vessels.

## 1. Introduction

The emergence of medical image segmentation as a novel biomedical image processing technology has significantly advanced the field of sustainable medical treatment [1,2]. Nowadays, retinal diseases such as diabetic retinopathy and glaucoma are widespread and pose a serious threat to people’s visual health [3]. Early diagnosis and regular eye examinations are key steps in the prevention and treatment of these diseases, and the application of retinal vascular segmentation technology is gaining increasing attention in this field, providing physicians with a precise and efficient tool to help detect and intervene earlier in these diseases, thereby protecting the visual health of patients. However, blood vessels are complex and varied globally, thin and fragile locally, and easy to fracture [4]. Therefore, automatic segmentation of blood vessels is a challenging task, especially for the accurate segmentation of thin blood vessels and their marginal parts [5]. The development of this technology will further promote the early diagnosis and treatment of retinal diseases and provide better vision protection strategies for patients [6].

Deep learning has revolutionized computer vision and machine learning. With the development of deep learning technology, automatic feature learning methods such as convolutional neural networks (CNNs) [7] and fully convolutional networks (FCNs) [8] have gradually replaced traditional manual feature extraction methods and have performed well in medical image segmentation tasks, including the segmentation of blood vessels. The U-Net proposed by Ronneberger et al. [9] in 2015 marked a significant advancement in deep learning in the field of medical image segmentation, which is characterized by a symmetric U-shaped structure that can effectively capture details and boundary information in the image. On the basis of U-Net, a series of new network structures continue to emerge, such as UNet++, V-Net, TransUNet, 3D U-Net, Axial-DeepLab, Deeplab v3+, R2UNet, refs. [10,11,12,13,14,15,16], etc. These networks further promote the development of medical image segmentation technology.

Inspired by U-Net and its derived models, researchers have proposed a variety of vessel segmentation methods. In 2019, Li et al. [17], committed to in-depth mining and the use of features from U-Net model to infer missing parts in images, rather than simply focusing on the improvement of U-Net model. IterNet can directly obtain detailed information from the processed results without looking back to the original image. However, the performance of the network on thin blood vessel segmentation task still needs to be improved. The RV-GAN network proposed by Kamran et al. [18], in 2021, consists of two generators and two discriminators. The generator segment the image by extracting local information such as small branches, while the generator tries to learn and preserve global information while generating less-detailed microvascular segmentation. However, RV-GANs require a lot of training time and computing resources. In 2023, Qi et al. [19] proposed dynamic snake convolution (DSConv) to enhance the perception of geometric structures by adaptive focusing on the thin and curved local features of tubular structures. In addition, DSCNet designed multi-view feature fusion strategies and continuity topology constraint loss to enhance the model’s perception of tubular structures. The multi-view feature fusion strategy can complement the multi-angle attention to features in the process of feature fusion to ensure the retention of important information from different global forms. The continuity topology constraint loss is based on persistent homology, so as to better constrain the topological continuity of segmentation.

Although many models have made great progress in blood vessel segmentation, there are still some problems: (1) the blood vessels are complex and variable globally, and it is difficult to extract thin vessels and their edge pixels in local thin and fragile vessels. (2) Stacking convolution layers to capture detailed features can lead to overfitting problems. To solve the above problems, this paper proposes an improved U-shaped network, which aims to improve the generalization ability of the model and extract the thin blood vessels and their boundary pixels as much as possible. The main contributions of this paper are as follows:(1)In view of the overall complex and variable structure of blood vessels and the local thin and fragile structure, this paper uses pyramid channel attention to capture detailed features more accurately, so as to extract more small blood vessels and their boundary pixels.(2)To solve the network overfitting problem, the convolutional block in this paper uses a pre-activated residual structure and adds the Dropblock module.(3)Experiments on three medical imaging datasets show that the proposed method has better performance on segmentation tasks than existing methods.

## 2. Methodology

In this section, a pyramid channel attention module is designed to focus on the feature information related to the structure of blood vessels so as to extract more small blood vessels and their boundary pixels. Meanwhile, convolutional blocks with pre-activated residual structures and Dropblock modules are used to alleviate the overfitting problem.

### 2.1. Datasets

The dataset DRIVE [20] consists of 40 color fundus images with a resolution of 584 × 564 pixels, derived from the Diabetic Retinopathy Screening Project in the Netherlands. Among them, there were 7 abnormal cases. The images were divided into a training set and a test set, each containing 20 images. In the training set, each image was manually segmented by an eye specialist. In the test set, each image was manually segmented by two different experts, and the segmentation result of the first expert was used as the performance evaluation criterion.

The dataset CHASE_DB1 [21] consists of 28 high-resolution fundus color images, each with a size of 999 × 960 pixels. These images were taken by specialized equipment to study and evaluate fundus blood vessel segmentation algorithms. In accordance with previous research [22,23], the first 20 images were used as a training set, while the remaining 8 images were used as a test set.

The dataset STARE [24] is a publicly available dataset for retinal vessel segmentation. This dataset consists of 20 color fundus images, each with a resolution of 700 × 605 pixels. Half of the images contained signs of lesions, and each image had the results of manual segmentation by two groups of experts. When dealing with the STARE data set, due to the lack of uniform partition, this paper adopts the 10-fold cross-validation method to carry out experiments. Specifically, this paper uses 18 images as training samples and the remaining images as test samples. This process is repeated 10 times to cover the entire data set, thereby minimizing bias and ensuring the reliability of the experimental results [25].

The details of the datasets preprocessing method are shown in Table 1.

### 2.2. Pyramid Channel Attention Module

The structure of the PCAM is shown in Figure 1a. In this paper, spatial pyramid pooling (SPP) [26] is first used to capture features at different scales. More comprehensive and rich information can be obtained through multi-scale maximum pooling of extracted features, thus improving the model’s perception ability of targets at different scales. The structure of SPP, as shown in Figure 1b [26], involves the use of four max-pooling layers with sizes of 2 × 2, 3 × 3, 5 × 5, and 6 × 6 to gather contextual information. Subsequently, the low-dimensional feature maps are upsampled, obtaining features of the same size as the original feature map through bilinear interpolation. This is conducted to better capture information at different scales.

The set of information descriptions obtained from SPP is used as input to CAM (as shown in Figure 1c) [27]. The channel attention module adaptively adjusts the features of channel dimensions so that the network can better focus on important channels and improve the model performance without increasing the computational cost too greatly. By learning the correlation between channels, CAM can highlight the channels related to blood vessels and thus better capture the characteristics of blood vessels. In addition, CAM can suppress channels unrelated to blood vessels, thus reducing noise and interference. Channel attention can be expressed as (1) [27], where F is the feature graph, δ represents the sigmoid function, $W_{0}\in R^{C/r\times C}$, $W_{1}\in R^{C\times C/r}$, and the MLP weights $W_{0}$ and $W_{1}$ are shared for both inputs.
(1)$$\begin{array}{cc} M_{c}\left(F\right) & =\delta (MLP(AvgPool(f))+MLP(MaxPool(F)))\\ \; & =\delta (W_{1}\left(W_{0}\left(F_{avg}^{c}\right)\right)+W_{1}\left(W_{0}\left(F_{max}^{c}\right)\right)) \end{array}$$

PCAM (as shown in Figure 1a) consists of SPP and CAM. First, the input feature graph F is operated by SPP to obtain the information description set S, which contains the multi-scale representation of the input features. Then, after the information description set S is operated by CAM, the obtained channel attention feature graph $M_{c}$ is multiplied with the information description set S element-by-element, and the generated feature graph is cascaded with the original feature graph F to form the pyramid channel attention feature graph $SM_{c}$. The process can be expressed as (2). This design enables the model to more accurately capture the characteristic information related to the blood vessel structure in the blood vessel segmentation task, thus improving the segmentation accuracy and robustness.
(2)$$\begin{array}{cc} SM_{c} & =\left[F,\;SPP(F)\times CAM(SPP(F))\right]\\ \; & =\left[F,\;S\times M_{c}\left(S\right)\right] \end{array}$$

### 2.3. Pre-Activated Residual Discard Convolution Block

#### 2.3.1. U-Net

U-Net performs well in the field of medical image segmentation, including retinal blood vessel segmentation, lung nodule segmentation, etc. In addition, it is also widely used in satellite image segmentation and industrial defect detection. U-Net contains a contracting path and an expansive path, forming an encoder–decoder structure. In the compression path, U-Net extracts the features of the image through a series of convolution and pooling operations. In the extended path, U-Net restores the spatial resolution of the image through up-sampling and skip-connection operations and retains the position information of the image, thus achieving pixel-level segmentation. U-Net is simple, efficient, understandable, easy to build, can be trained from small data sets, and its network structure allows it to efficiently acquire contextual and positional information about images.

The proportion of foreground pixels in retinal blood vessel images is much smaller than that of background pixels, which means that U-Net faces the problem of category imbalance during training, making it easier for the network to predict non-vascular regions and ignore blood vessels. In addition, retinal blood vessels show a long, convoluted, and intricate distribution pattern, and their intersections are complicated and easy to fracture. For this kind of structure, the accuracy of spatial information is very important. However, in U-Net, continuous upsampling operations may result in loss of spatial information. Although U-Net uses skip connections to preserve some low-level features, it cannot completely avoid information loss. The residual block in ResNet [28] has a similar idea to a skip connection, which is to improve the performance of deep neural networks by retaining low-level features.

#### 2.3.2. Residual Unit

In order to solve the problem of network degradation and gradient disappearance in deep neural networks, He et al. [28] introduced a residual module into the network, adding a direct connection path between input and output to directly perform the mapping. In this way, the network only needs to learn the residual, which effectively solves the problem of network degradation. The general forms of the convolution block in U-Net (as shown in Figure 2a) [9] and the residual unit in ResNet (as shown in Figure 2b) [28] can be expressed as Formulas (3) [9] and (4) [28], respectively.
(3)$$y_{1}=F\left(x_{1}\right)$$
(4)$$y_{1}=x_{1}+F\left(x_{1}\right)$$
where $x_{1}$ and $y_{1}$ are the input and output of the *l*-th convolution unit, and F(·) represents the convolution operation.

On this basis, Zhang et al. [29] applied this idea to road extraction and achieved remarkable results. It is worth noting that road images and blood vessel images have similar features, that is, the pixel to be extracted occupies a large area in the entire image, but the proportion is very small. Therefore, we believe that a residual unit would also have a positive impact on blood vessel segmentation.

#### 2.3.3. Dropblock

In the field of medical image segmentation, the available data sets are limited and the training data are small. Even with the use of data enhancement techniques, the standard U-Net model is prone to overfitting when applied directly to such tasks. Dropout [30] can effectively reduce overfitting in the fully connected layer. The core idea is to randomly “drop” some neurons during training, that is, set the output to “0”, thereby reducing the model’s dependence on some specific neurons and preventing overfitting. However, in the convolutional layer, the connections between neurons are local, and applying standard Dropout directly to the convolutional layer may disrupt the continuity of the feature map, resulting in information loss.

Inspired by [31,32,33], this paper uses Dropblock to regularize networks with residual connections and improves the generalization ability of neural networks by randomly masking continuous regions in convolutional feature graphs. The main difference between Dropblock and Dropout is that Dropblock randomly drops an area of a specific size during training, rather than a single neuron, which preserves the continuity of the feature map and avoids excessive destruction of features. Figure 3a,b [31] illustrate the function diagrams of Dropout and Dropblock, respectively. Dropblock is characterized by two main parameters: β and ρ. Here, β represents the size of the drop area; when β=1, Dropblock simplifies to general Dropout. Additionally, ρ controls the number of dropped features as defined in Formula (5) [31].
(5)$$\rho =\frac{\left(1-k_{eep\_prob}\right)\times \left(\omega \times h\right)}{\beta^{2}\times \left(\omega -\beta +1\right)\times \left(h-\beta +1\right)}$$
where, $k_{eep\_prob}$ is the probability of neuron inactivation, w and h are the width and height of the feature map, respectively. The size of the effective region of the feature map is (ω−β+1)×(h−β+1). Therefore, Dropblock can effectively prevent network overfitting and improve the generalization ability of the model.

In addition, this paper placed BN and ReLU in front of the convolutional layer to form the pre-activation mode [28] and formed q pre-activated residual discard convolution block (PRDC), which further enhanced the regularization of the network and accelerated the training speed and convergence ability of the network (as shown in Figure 2c). The architecture of the model proposed in this paper is shown in Figure 4.

## 3. Experiments

### 3.1. Evaluation Metrics

In this paper, intersection over union (IOU), accuracy (Acc), sensitivity (Sen), specificity (Spe) and F1 score (F1) were used to evaluate the performance of the binary segmentation model, and their definitions were as follows: (6)$$IOU=\frac{TP}{TP+FP+FN}$$(7)$$Acc=\frac{TP+TN}{TP+TN+FP+FN}$$(8)$$Sen=\frac{TP}{TP+FN}$$(9)$$Spe=\frac{TN}{TN+FP}$$(10)$$F1=\frac{2TP}{2TP+FP+FN}$$

False negatives (FN), true positives (TP), true negatives (TN), and false positives (FP) are the four basic elements in calculating metrics. Here, true positives (TP) and true negatives (TN) represent the number of correctly segmented vascular pixels and non-vascular pixels, respectively. False positives (FP) and false negatives (FN) represent the number of missegmented vascular and non-vascular pixels, respectively.

### 3.2. Implementation Details

All experiments in this paper are carried out in Windows 10, and the GPU used was an RTX 3080. The dataset DRIVE provides an official mask for the test dataset but datasets CHASE_DB1 and STARE do not. In order to make a fair comparison, this paper generates a FoV mask through simple color threshold processing on the original image (as shown in Figure 5) [17]. In the process of training, this paper uses data enhancement to increase the diversity of data. During the test, the image was divided into patches for prediction to reduce the computing and memory requirements.

## 4. Results

### 4.1. Comparison Studies

This paper compares some of the most advanced models on three datasets: DRIVE, CHASE_DB1, and STARE. These models include U-Net, U-Net++, Attention U-Net, IterNet, RV-GAN, etc. It should be noted that only U-Net, U-Net++, and Attention U-Net were tested in this paper, while the results of other models were obtained from the corresponding papers. The same training technique was used for each model in the repetition test. Table 2 and Table 3 list the qualitative results of each model. It can be seen from the table that the model in this paper has achieved good results on the whole, and its Sen, F1 and IOU scores have been improved to some extent. Additionally, it is worth mentioning that the model in this paper achieved significantly higher Sen scores on CHASE_DB1 compared to other methods. Many studies [34,35] have pointed out that an increase in Sen score is positively correlated with the model’s ability to extract microvessels and boundary pixels more efficiently. Therefore, the model in this paper has a relatively strong ability in microvascular structure extraction.

Specifically, in the datasets DRIVE, CHASE_DB1, and STARE, compared with popular methods such as U-Net, U-Net++, and Attention U-Net, the Sen, F1 and IOU of the model presented in this paper are all improved. Compared with U-Net, the Sen scores of the model in this paper are increased by 7.12%, 9.65%, and 5.36% on the three datasets, respectively. Compared with U-Net++, Sen scores were increased by 5.63%, 2.05%, and 3.02% on the three datasets, respectively. Compared with Attention U-Net, the Sen scores were increased by 3.37%, 5.67%, and 4.88% on the three datasets, respectively. Compared with RV-GAN model, the Sen scores of DRIVE and CHASE_DB1 improved by 2.2% and 4.07%, respectively. However, in STARE, the Sen score of the RV-GAN model is higher than that of the model in this paper. However, it is worth mentioning that the model in this paper has the highest IOU score on all three datasets.

Among these three datasets, the Spe scores of our model are slightly lower compared to other models. This is because PCAM and PRDC incorporated in this paper make the model pay more attention to thin blood vessels and their edge pixels, which will lead to certain false positive problems, that is, non-vascular pixels are predicted to be vascular pixels. This phenomenon can be qualitatively expressed as an increase in its FP score. According to Equation (Equation 6), an increase in FP will lead to a decrease in its Spe score. Similarly, its Acc score will be slightly lower than other models.

In this paper, the results are further visualized, and the blood vessel segmentation results of U-Net, U-Net++, Attention U-Net and the proposed model are included in Figure 6. Retinal blood vessels are complex, thin, and fragile, so this paper enlarges the details to be able to observe their segmentation results more intuitively. Taking STARE as an example, regardless of whether the box is red or blue, the model in this paper is able to extract more thin blood vessels and edge pixels, and also outperforms other models in terms of connectivity. In conclusion, the model in this paper has a strong ability to extract microvascular structure and has excellent performance in the task of blood vessel segmentation.

### 4.2. Ablation Studies

As shown in Figure 4, the model presented in this paper can be regarded as an encoder–decoder network consisting of a pyramid channel attention module (PCAM, as shown in Figure 1a) and a pre-activated residual discard convolution block (PRDC, as shown in Figure 2c). In order to verify the effectiveness of the components, this paper conducts ablation experiments on the model to evaluate the influence of each component on the final segmentation result. Table 4, Table 5 and Table 6 show the results of different combinations for DRIVE, CHASE_DB1, and STARE, respectively. In this paper, U-Net is set to the baseline. “baseline+PCAM” represents the integration of pyramid channel attention into the encoder. “baseline+PRDC” indicates a replacement of the original convolution block in the baseline with the pre-activated residual discard convolution block. “baseline+PCAM+PRDC” represents the integration of the pre-activated residual discard convolution block and pyramid channel attention into the baseline at the same time. All experiments are carried out using the same hyperparameter configuration. This paper takes the dataset DRIVE as an example for specific analysis.

By incorporating pyramid channel attention into the baseline (“baseline+PCAM”), the Acc, Sen, F1, and IOU scores improved by 0.04%, 2.79%, 0.7%, and 0.99%, respectively. This indicates that integrating pyramid channel attention allows the model to focus more on features related to vascular structures. After replacing the original convolution block with the pre-activated residual discard convolution block, its Acc, Sen, F1, IOU scores are increased by 0.18%, 4.3%, 1.46%, 2.08%, respectively. This shows that incorporating residual convolution that focuses on overfitting can improve model performance. Finally, this paper further integrated PCAM and PRDC into the baseline (“baseline+PCAM+PRDC”) to verify the additive effect of components. This method obtained the highest Sen, F1 and IOU values in the ablation experiment, indicating that the model in this paper has strong ability in microvascular structure extraction. Its Spe score is the lowest in ablation experiments, and the reasons have been analyzed in detail.

## 5. Conclusions

In this paper, a U-shaped network based on a pyramidal attention mechanism is proposed to solve the vascular segmentation problem. In this paper, a pyramid channel attention module is introduced into the encoder. The pyramid structure of the module allows the model to focus on features of different scales at the same time. By combining the channel attention mechanism, the model can pay more attention to the channels related to blood vessels, thereby enhancing the focus on blood vessel structure and suppressing features unrelated to blood vessels. This design improves the sensitivity and precision of the vascular segmentation task, allowing it to effectively predict complex, variable, thin, and fragile vascular structures. Furthermore, in this paper, we replaced the standard convolutional blocks with residual-connected blocks. Within these blocks, we introduced the Dropblock module and employed a pre-activation mode to effectively mitigate overfitting issues. Experiments show that the model in this paper has strong ability in extracting thin vessel structures. This shows that the combination of the pyramid channel attention module and the pre-activated residual discard convolution block can effectively improve the performance and generalization ability of the model. In the future, we will further apply the network to 3D vascular datasets, such as magnetic resonance angiography and computed tomography angiography.

## Figures and Tables

**Figure 1 sensors-24-04014-f001:**
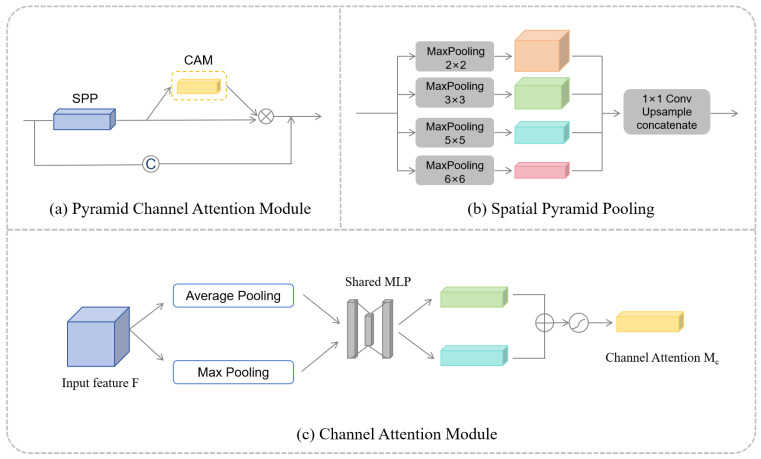
Pyramid channel attention module and submodule.

**Figure 2 sensors-24-04014-f002:**
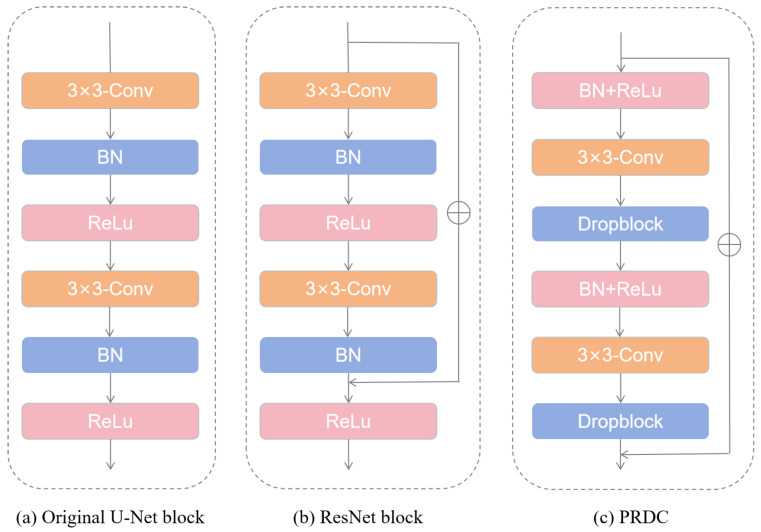
Convolutional block.

**Figure 3 sensors-24-04014-f003:**
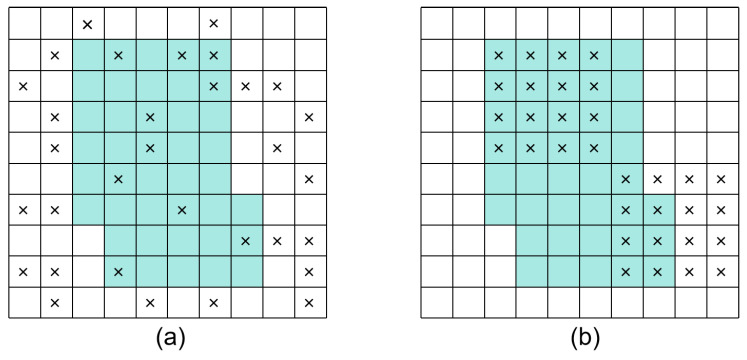
Dropout (**a**) and Dropblock (**b**) function diagrams.

**Figure 4 sensors-24-04014-f004:**
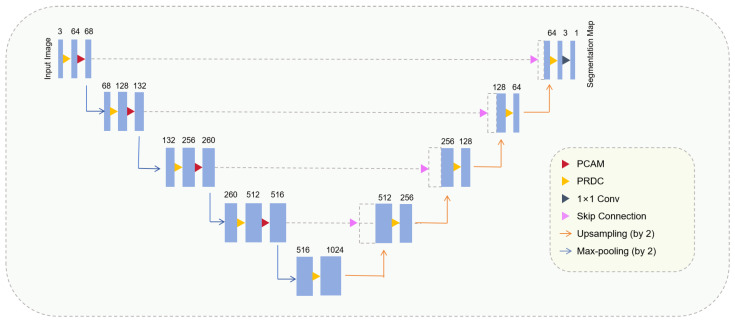
The proposed algorithm structure diagram.

**Figure 5 sensors-24-04014-f005:**
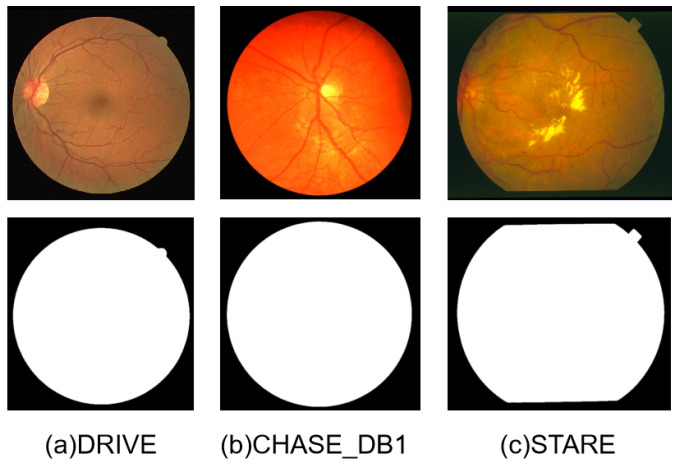
Raw images and masks from the datasets.

**Figure 6 sensors-24-04014-f006:**
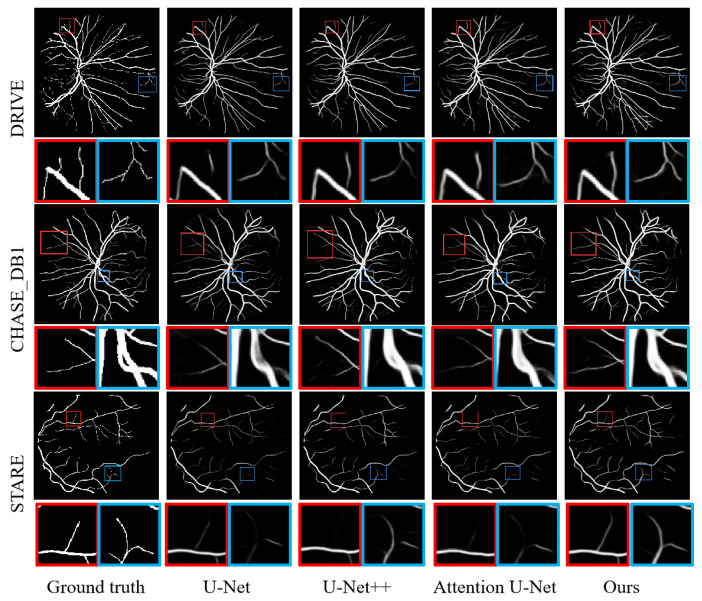
Visualization of vessel segmentation results. This paper uses red and blue boxes to highlight the details and zoom in to make the results more intuitive.

**Table 1 sensors-24-04014-t001:** The specific information of DRIVE, CHASE_DB1 and STARE.

Datasets	DRIVE	CHASE_DB1	STARE
Total number	40	28	20
Train/Test number	20/20	20/8	18/2
Resolution (pixel)	565 × 584	999 × 960	700 × 605
Resize (pixel)	576 × 576	960 × 960	592 × 592
Augmentation methods	Randomly adjust brightness, contrast, saturation; Random rotation, flip, shift, zoom		

**Table 2 sensors-24-04014-t002:** Comparison results of different model experiments on DRIVE and CHASE_DB1. **Bold** values indicate the highest scores achieved for the respective metrics.

Methods	Times	DRIVE					CHASE_DB1				
		Acc	Sen	Spe	F1	IOU	Acc	Sen	Spe	F1	IOU
U-Net [9]	2015	0.9554	0.7435	0.9860	0.8078	0.6776	0.9638	0.7641	0.9850	0.8018	0.6692
U-Net++ [10]	2018	0.9552	0.7584	0.9837	0.8104	0.6812	0.9654	0.8401	0.9787	0.8232	0.6995
Attention U-Net [36]	2018	0.9560	0.7810	0.9812	0.8172	0.6910	0.9647	0.8039	0.9817	0.8136	0.6858
AG-Net [22]	2019	0.9692	0.8100	0.9848	/	0.6965	**0.9743**	0.8186	0.9848	/	0.6669
IterNet [17]	2019	0.9573	0.7735	0.9838	0.8205	/	0.9655	0.7970	0.9823	0.8073	/
RVSeg-Net [23]	2020	0.9681	0.8107	0.9845	/	/	0.9726	0.8069	0.9836	/	/
VSSC Net [37]	2021	0.9627	0.7827	0.9821	/	/	0.9633	0.7233	**0.9865**	/	/
RV-GAN [18]	2021	**0.9790**	0.7927	**0.9969**	**0.8690**	/	0.9697	0.8199	0.9806	**0.8957**	/
Ours	2024	0.9570	**0.8147**	0.9776	0.8271	**0.7052**	0.9646	**0.8606**	0.9757	0.8235	**0.6999**

**Table 3 sensors-24-04014-t003:** Comparison results of different model experiments on STARE. **Bold** values indicate the highest scores achieved for the respective metrics.

Methods	Times	STARE				
		Acc	Sen	Spe	F1	IOU
U-Net [9]	2015	0.9650	0.7595	0.9882	0.8133	0.6884
U-Net++ [10]	2018	0.9667	0.7829	0.9878	0.8264	0.7063
Attention U-Net [36]	2018	0.9654	0.7643	0.9883	0.8163	0.6922
IterNet [17]	2019	0.9701	0.7715	**0.9886**	0.8146	/
RV-GAN [18]	2021	**0.9754**	**0.8356**	0.9864	0.8223	/
Ours	2024	0.9657	0.8131	0.9832	**0.8279**	**0.7077**

**Table 4 sensors-24-04014-t004:** Comparison of different combinations on DRIVE. **Bold** values indicate the highest scores achieved for the respective metrics.

Methods	DRIVE				
	Acc	Sen	Spe	F1	IOU
baseline	0.9554	0.7435	**0.9860**	0.8078	0.6776
baseline+PCAM	0.9558	0.7714	0.9824	0.8148	0.6875
baseline+PRDC	**0.9572**	0.7865	0.9818	0.8224	0.6984
baseline+PCAM+PRDC	0.9570	**0.8147**	0.9776	**0.8271**	**0.7052**

**Table 5 sensors-24-04014-t005:** Comparison of different combinations on CHASE_DB1. **Bold** values indicate the highest scores achieved for the respective metrics.

Methods	CHASE_DB1				
	Acc	Sen	Spe	F1	IOU
baseline	0.9638	0.7641	**0.9850**	0.8018	0.6692
baseline+PCAM	0.9632	0.8099	0.9795	0.8084	0.6784
baseline+PRDC	**0.9653**	0.8452	0.9780	0.8234	0.6998
baseline+PCAM+PRDC	0.9646	**0.8606**	0.9757	**0.8235**	**0.6999**

**Table 6 sensors-24-04014-t006:** Comparison of different combinations on STARE. **Bold** values indicate the highest scores achieved for the respective metrics.

Methods	STARE				
	Acc	Sen	Spe	F1	IOU
baseline	0.9650	0.7595	**0.9882**	0.8133	0.6884
baseline+PCAM	0.9634	0.7868	0.9837	0.8135	0.6874
baseline+PRDC	**0.9671**	0.8020	0.9861	**0.8321**	**0.7140**
baseline+PCAM+PRDC	0.9657	**0.8131**	0.9832	0.8279	0.7077

## Data Availability

The data presented in this study are available on request from the corresponding author. The datasets we use are public datasets. https://drive.grand-challenge.org/Download/, https://blogs.kingston.ac.uk/retinal/chasedb1/, https://paperswithcode.com/dataset/stare (all accessed on 22 May 2024).

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
