# Peer review of "A Microvascular Segmentation Network Based on Pyramidal Attention Mechanism"

_sensors, 2024, doi:10.3390/s24124014_

Round 1

Reviewer 1 Report

Comments and Suggestions for Authors

Dear Authors 

Review for the paper titled: “Microvascular Segmentation Network Based on Pyramidal Attention Mechanism”

The paper looks good in terms of structure and organization. However, in terms of content the following points need to be considered:

1.      A limited literature review is presented in the paper; more literature review needs to be conducted on the area. In addition, to the comparison table that needs to be added at the end of the literature to highlight the main gaps and challenges within the research area.

2.      Methodology section need to be reorganized such that authors need to start from the datasets description, methodology used in preprocessing datasets (images), then model development and evaluation.

3.      Why the authors decided to use Dropblock modules to alleviate the overfitting problem??

4.      In section 2.1 the authors started the section by explaining Figure 1 (c) not Figure 1 (a)???

5.      Authors should consider adding references for Equations 1 to 4 in the paper.

6.      Authors should add references for Figures 1 to 3 if they obtained them from other references (open access source) but if they generated them it is fine to keep them as they are.

All the best 

Reviewer 2 Report

Comments and Suggestions for Authors

The paper describes using of a pyramid channel attention module in a U-net structure with pre-activated residual discard convolution block for improving the pixel-wise segmentation of retinal microvessel dataset. 

I find the method and description clear with minor edits suggestion. 

page1 line 29 CNN& FCN should be spell out the first time for the interested of out of the field general readers.  page 2 line 50 is DSC mean DSConv?   page 9 table 6  the highest data in F1 and IOU is not bolded instead of the baseline + PCAM+PRDc. s there a reason?   Comments on the Quality of English Language

none
